# IPSC-Derived Astrocytes Contribute to In Vitro Modeling of Parkinson’s Disease Caused by the *GBA1* N370S Mutation

**DOI:** 10.3390/ijms25010327

**Published:** 2023-12-26

**Authors:** Elena S. Yarkova, Elena V. Grigor’eva, Sergey P. Medvedev, Sophia V. Pavlova, Suren M. Zakian, Anastasia A. Malakhova

**Affiliations:** 1Department of Natural Sciences, Novosibirsk State University, 630090 Novosibirsk, Russia; e.drozdova2@g.nsu.ru; 2Institute of Cytology and Genetics, Siberian Branch of Russian Academy of Sciences, 630090 Novosibirsk, Russia; medvedev@bionet.nsc.ru (S.P.M.); spav@bionet.nsc.ru (S.V.P.); zakian@bionet.nsc.ru (S.M.Z.); amal@bionet.nsc.ru (A.A.M.); 3Institute of Chemical Biology and Fundamental Medicine, Siberian Branch of Russian Academy of Sciences, 630090 Novosibirsk, Russia; 4Meshalkin National Medical Research Center, Ministry of Health of the Russian Federation, 630055 Novosibirsk, Russia

**Keywords:** induced pluripotent stem cells, *GBA1*, Parkinson’s disease, astrocytes

## Abstract

Parkinson’s disease (PD) is a neurodegenerative disorder that ranks second in prevalence after Alzheimer’s disease. The number of PD diagnoses increases annually. Nevertheless, modern PD treatments merely mitigate symptoms rather than preventing neurodegeneration progression. The creation of an appropriate model to thoroughly study the mechanisms of PD pathogenesis remains a current challenge in biomedicine. Recently, there has been an increase in data regarding the involvement of not only dopaminergic neurons of the substantia nigra but also astrocytes in the pathogenesis of PD. Cell models based on induced pluripotent stem cells (iPSCs) and their differentiated derivatives are a useful tool for studying the contribution and interaction of these two cell types in PD. Here, we generated two iPSC lines, ICGi034-B and ICGi034-C, by reprogramming peripheral blood mononuclear cells of a patient with a heterozygous mutation c.1226A>G (p.N370S) in the *GBA1* gene by non-integrating episomal vectors encoding *OCT4*, *KLF4*, *L-MYC*, *SOX2*, *LIN28*, and *mp53DD*. The iPSC lines demonstrate the expression of pluripotency markers and are capable of differentiating into three germ layers. We differentiated the ICGi034-B and ICGi034-C iPSC lines into astrocytes. This resulting cell model can be used to study the involvement of astrocytes in the pathogenesis of GBA-associated PD.

## 1. Introduction

Parkinson’s disease (PD) is a neurodegenerative disorder in which approximately 5% of cases are hereditary [1]. The most common genetic risk factor for PD is a mutated *GBA1* gene [2,3]. About 400 different mutations in the *GBA1* gene are known, some of which have pathogenic status [4,5]. The most common are the single nucleotide substitutions p.N370S and p.L444P [6], which in the heterozygous state increase the risk of developing PD by a factor of 6–10 [5].

The product of the *GBA1* gene is the lysosomal enzyme ꞵ-glucocerebrosidase (GCase). GCase is critical for many cell types, including neurons and astrocytes [7,8]. While dopaminergic neurons of the substantia nigra are the main cell type to die in PD, attention has recently been focused on astrocytes because their reactive state can significantly influence the processes of neuroinflammation and neurodegeneration [9]. Some studies show that GCase dysfunction resulting from mutations in the *GBA1* gene causes more severe damage in astrocytes than in neurons. For example, mitochondrial fragmentation is more severe in astrocytes [7]. Thus, mutations in the *GBA1* gene can cause significant disruptions in the intracellular processes of astrocytes and pose an additional threat to the survival of dopaminergic neurons [10].

It is still not clear what mechanisms cause the development of GBA-associated PD and how to stop the progressive neurodegeneration in the substantia nigra. Understanding the pathogenesis is the key to finding a cure for this pathology. Previously, we obtained dopaminergic neurons by directed differentiation of induced pluripotent stem cells (iPSCs) from patients with PD associated with the pathogenic mutation N370S in the *GBA1* gene. We observed a decrease in GBA activity and altered expression of lysosomal enzymes compared to healthy controls [11]. In this work, we generate an iPSC-based cell model to examine the role of astrocytes in GBA-associated PD pathogenesis.

## 2. Results

### 2.1. ICGi034-B and ICGi034-C Cell Lines Have a Typical Morphology for Induced Pluripotent Stem Cells (iPSCs) and Are AP-Positive

A mononuclear fraction was isolated from the peripheral blood of a patient who carries the heterozygous single nucleotide polymorphism rs76763715 (c.1226A>G, p.N370S) in the *GBA1* gene. The resulting peripheral blood mononuclear cells (PBMCs) were transfected with episomal vectors expressing reprogramming factors: *OCT4*, *KLF4*, *L-MYC*, *SOX2*, *LIN28*, and *Trp53* [12]. The obtained iPSC lines were registered in the Human Pluripotent Stem Cell Registry, hPSCreg, under the names ICGi034-B (https://hpscreg.eu/cell-line/ICGi034-B, accessed on 1 December 2023), ICGi034-C (https://hpscreg.eu/cell-line/ICGi034-C, accessed on 1 December 2023). One of the iPSC lines (ICGi034-A) was characterized earlier [13]; this article presents a detailed characterization of the other two lines—ICGi034-B and ICGi034-C (Table 1). Colonies of cells are round and flat, which corresponds to the typical morphology of human pluripotent stem cells (Figure 1A). In addition, there is alkaline phosphatase (AP) activity in the colonies (Figure 1B).

### 2.2. Lines ICGi034-B and ICGi034-C Demonstrate Expression of Pluripotency Markers

Immunofluorescence analysis demonstrates the expression of pluripotency markers at passage 5, such as the transcription factors OCT4 and SOX2, as well as the surface antigens SSEA-4 and TRA-1-60 (Figure 1E). In both cell lines, high levels of expression of the *OCT4*, *SOX2*, and *NANOG* genes are observed by quantitative PCR (Figure 1C). Expression is comparable to that for the human embryonic stem cell line HUES9 (HVRDe009-A), which we used as a positive control [14].

### 2.3. Lines Are Capable of Differentiating into Derivatives of Three Germ Layers

The most accurate confirmation of the pluripotent status of the ICGi034-B and ICGi034-C lines was made by a test for spontaneous differentiation in vitro in embryoid bodies. The experiment demonstrated that the ICGi034-B and ICGi034-C lines could differentiate into three germ layer derivatives. The expression of ectoderm markers (tubulin β 3 (TUBB3/TUJ1), methionine aminopeptidase 2 (MAP2)), mesoderm markers (α-smooth muscle actin (αSMA), NK2 Homeobox 5 (NKX2.5)), and endoderm markers (alpha-fetoprotein (AFP), hepatocyte nuclear factor 3 beta (HNF3β/FOXA2)) was revealed by immunofluorescent staining (Figure 1F).

### 2.4. Lines ICGi034-B and ICGi034-C Have a Normal Karyotype and Are Genetically Identical to the Parental Peripheral Blood Mononuclear Cells (PBMCs)

Analysis of 50 metaphase plates demonstrated that ICGi034-B and ICGi034-C had a normal karyotype at passage 8 (46,XX; Figure 1D). The genetic identity of the resulting iPSC lines to the original PBMCs was demonstrated at 25 polymorphic loci using short tandem repeat (STR) analysis. The single nucleotide polymorphism rs76763715 (c.1226A>G, p.N370S) in the *GBA1* gene in ICGi034-B and ICGi034-C, as well as in the original PBMCs, was detected using Sanger sequencing (Figure 1G).

### 2.5. ICGi034-B and ICGi034-C Lines Have Successfully Passed Additional Quality Control Tests

It was also shown that the elimination of episomal vectors occurred at passage 15 (ICGi034-B) and passage 16 (ICGi034-C) (Appendix A). Testing for mycoplasma contamination using PCR revealed that the ICGi034-B and ICGi034-C lines were free of mycoplasma infection (Appendix A).

### 2.6. IPSCs ICGi034-B and ICGi034-C Successfully Differentiate into Astrocytes

Twenty-four hours before the start of the directed differentiation protocol, iPSCs were seeded in a dense monolayer (80–100%) on Matrigel-GFR in the absence of pluripotency maintenance factors. Directed differentiation was carried out according to the protocol published previously [15], with modifications (Figure 2A).

Until day 11, the protocol is aimed at obtaining neural stem cells. Stimulation of differentiation in the neuroectodermal direction is carried out owing to dual SMAD inhibition [16], which is achieved by adding small molecules LDN193189 and SB431542 to the culture medium. These substances inhibit the expression of pluripotency genes and also suppress differentiation in the mesodermal and endodermal directions. The factors CHIR99021, Sonic Hedgehog (SHH), and purmorphamine are required to direct differentiation toward midbrain progenitors.

After 11 days of differentiation, cell lines were transferred onto Matrigel-GFR with the addition of a ROCK inhibitor for the first 24 h. On the 13th day of differentiation, the growth factors FGF, EGF, and CNTF that are necessary to obtain mature astrocytes were added to the culture medium (Figure 2A). Throughout the process of terminal differentiation, the cells were passaged every 3–4 days. Some neurons that emerged during differentiation failed to survive the passaging. This enables the acquisition of a pure astrocyte culture free of neuron contamination.

On day 103 of the differentiation process, the cells begin to acquire the characteristic stellate shape of astrocytes and express astroglial markers, glial fibrillary acidic protein (GFAP), and brain natriuretic peptide (S100β) (Figure 2B,C).

## 3. Discussion

PD is a common neurodegenerative disease affecting a growing proportion of the able-bodied population [17] and becoming more prevalent as life expectancy increases in developed countries. PD is characterized by selective degeneration of dopaminergic neurons of the substantia nigra pars compacta [18]. This leads to disruption of the dopaminergic pathway to the basal ganglia. Currently, an excessive proinflammatory reaction in astroglial cells is considered one of the mechanisms of neuronal damage [9,18].

However, modern treatments for PD do not prevent the progression of neurodegeneration but only relieve symptoms [19]. It is important to create adequate models for a detailed study of PD pathogenesis mechanisms. A unique platform to study the mechanisms of early disease development and to test potential drugs are cell models based on neural derivatives of iPSCs obtained from patients with hereditary forms of PD. Previously, we studied the activity of GCase and other lysosomal enzymes in dopaminergic neurons differentiated from iPSCs of patients with *GBA1* mutations. We also examined the efficacy of molecular chaperones in disease pathogenesis [11,20]. Here, we obtained and characterized two iPSC lines of a patient with pathogenic genetic variant N370S in the *GBA1* gene. We also demonstrated an effective differentiation of the iPSC lines into astrocytes.

Astrocytes are the most common type of glial cells in the human brain [21]. They are critically necessary for maintaining the vital activity of neurons. For example, astrocytes produce glial-derived neurotrophic factor (GDNF), which is important for the development and survival of dopaminergic neurons [7]. Regulation of synaptic transmission, metabolic support, and prevention of the spread of toxic signals are also the merits of astrocytes [22].

Like neurons, astrocytes express PD-related genes, such as *PARK2*, *PINK1*, *DJ-1*, *LRRK2*, and *GBA1*. Mutations in these genes have been shown to cause astrocyte dysfunction [7]. In particular, the p.N370S mutation in the *GBA1* gene causes a decrease in the function of lysosomal ꞵ-glucocerebrosidase [4,5]. The mutant enzyme is unable to degrade its substrate, lysosphingolipids, which accumulate in the lysosome and lead to lysosomal dysfunction. As a result, the toxic forms of α-synuclein protein are accumulated [23]. The presence of α-synuclein protein has been observed in both neurons and astrocytes [8,24]. It has been shown that astrocytes not only suffer from the effects of neurotoxic α-synuclein but also participate in its distribution in the brain [25]. Research has demonstrated that lysosomal dysfunction in astrocytes results in the accumulation of α-synuclein and contributes to astrocyte-induced inflammation [26]

Pathological changes in cells caused by expression of the defective GCase enzyme lead to disruption of the protein processing system and oxidative stress [27]. The study of the processes of oxidative stress and endoplasmic reticulum stress in cells is possible by means of genetically encoded biosensors [28]. The sequences of transgenes encoding biosensors can be introduced into the cell using the CRISPR/Cas9 genome editing system. Thus, the resulting cellular model based on neural derivatives of iPSCs with a mutation in the *GBA1* gene is a unique tool for studying the pathological processes occurring in the cell during the development of PD.

Studies on the generation of PD models based on astrocytes derived from iPSCs and their successful application to study various aspects of PD pathogenesis are increasingly appearing in the literature [29,30,31]. However, it is important to note that the model has some peculiarities because of astroglial cell ontogenesis. This type of cell originates from various progenitor cells, resulting in a heterogeneous population of mature astrocytes, even in the culture of astrocytes obtained from iPSCs [32]. Additionally, terminally differentiated astroglia exhibit plasticity and can transition into various reactive states, which can have diametrically opposite effects, either neuropreservative or neurotoxic [10]. However, the heterogeneity and reactivity of astrocytes in vitro may also reflect the processes occurring in the human brain. This brings our research closer to the in vivo system. The cell model created in our study enables the investigation of precise intracellular mechanisms and brings us closer to understanding the causes of neurodegeneration in PD. Our study provides an opportunity to investigate the contribution of astrocytes and dopaminergic neurons to the pathogenesis of PD associated with a mutation in the *GBA1* gene by co-culturing these two types of neural derivatives.

## 4. Materials and Methods

The study was approved by the Research Ethics Committee of the Federal Neurosurgical Center (Novosibirsk, Russia), protocol No. 1, dated 14 March 2017. PBMCs of a patient with PD were provided by the Federal Neurosurgical Center (Novosibirsk, Russia). The patient signed the informed consent form.

### 4.1. Obtaining and Culturing iPSCs

The methods used to obtain and characterize iPSC lines in detail are described in [11]. Briefly, nucleofection of PBMCs was carried out using a set of episomal vectors (ID Addgene #41855–58, #41813–14) on a Neon Transfection System (Thermo Fisher Scientific, Waltham, MA, USA).

iPSCs were cultured on mitotically inactivated mouse embryonic fibroblasts in following medium: 82% KnockOut DMEM medium, 15% KoSR, 1% Gluta-MAX, 1% Pen-Strep, 1% MEM NEAA (all Thermo Fisher Scientific, Waltham, MA, USA), 0.1 mM β-Mercaptoethanol (Sigma-Aldrich, Darmstadt, Germany), 10 ng/mL basic FGF (SCI Store, Moscow, Russia).

iPSC passaging was performed using TrypLE Express (Thermo Fisher Scientific, Waltham, MA, USA). Cells were split 1:10 in the iPSC medium with the addition of 2 µM Thiazovivin (ROCK inhibitor, Sigma-Aldrich, Darmstadt, Germany) for the first 24 h.

### 4.2. DNA Isolation

DNA was isolated using Quick-DNA Miniprep Kit (Zymo Research, Irvine, CA, USA) for STR analysis or extracted by QuickExtract™ DNA Extraction Solution (Lucigen, Madison, WI, USA) for *GBA1* gene mutation analysis, episome, and mycoplasma detection.

### 4.3. Detection of Mycoplasma and EBNA

The presence of mycoplasma and EBNA was checked by PCR (95 °C, 5 min; 35 cycles: 95 °C, 15 s, 60 °C, 15 s, 72 °C, 20 s) using the primers listed in Table 2 [33].

### 4.4. Quantitative PCR

Cells were placed in a conical tube and lysed using Trizol Reagent. Reverse transcription of RNA was performed using M-MuLV reverse transcriptase (Biolabmix, Novosibirsk, Russia). Quantitative PCR (qPCR) was performed in a LightCycler 480 II system (Roche, Basel, Switzerland) with a BioMaster HS-qPCR SYBR Blue 2× kit (Biolabmix, Novosibirsk, Russia) according to the following program: 95 °C 5 min; 40 cycles: 95 °C 10 s, 60 °C 1 min. CT values were normalized to beta-2-microglobulin using the ΔΔCT method.

### 4.5. Sanger Sequencing

Sanger sequencing was used to confirm the single nucleotide polymorphism rs76763715 (c.1226A>G, p.N370S) in the *GBA1* gene. The primers used are presented in Table 2. Reactions were carried out on a T100 thermal cycler (Bio-Rad, Hercules, CA, USA) using BioMaster HS-Taq PCR-Color (2×) (Biolabmix, Novosibirsk, Russia) according to the program: 95 °C, 3 min; further 35 cycles: 95 °C, 30 s; 60 °C, 30 s; 72 °C, 30 s; and 72 °C, 5 min. Next, we used BigDye Terminator V. 3.1. Cycle Sequencing Kit (Applied Biosystems, Austin, TX, USA). Sanger sequencing reactions were analyzed on an ABI 3130XL genetic analyzer at the Genomics Center of the SB RAS (http://www.niboch.nsc.ru/doku.php/corefacility, accessed on 1 December 2023).

### 4.6. Karyotypic Analysis

Karyotype analysis was performed as described earlier [34]. G-banding and karyotyping were carried out in the Institute of Medical Genetics of the Tomsk National Research Medical Center of the Russian Academy of Sciences using the International System of Human Cytogenetic Nomenclature.

### 4.7. Spontaneous Differentiation In Vitro

The iPSCs were treated with 0.15% collagenase type IV (Thermo Fisher Scientific, Waltham, MA, USA) and plated onto 35 mm dishes coated with 1% gelatin in iPSC culture medium without the addition of bFGF. Cells were cultured for 9–14 days until embryoid bodies were formed, which were then transferred onto Matrigel-coated plates and cultured for another week. Next, immunofluorescence analysis was performed to detect markers of the three germ layers. The list of antibodies is presented in Table 2.

### 4.8. Immunofluorescence Analysis

Immunofluorescent staining was performed according to the previously described procedure [34]. Briefly, iPSCs were fixed in 4% paraformaldehyde for 10 min, then permeabilized in 0.5% Triton-X100 for 30 min, and incubated with a blocking buffer (1% BSA in PBS) for 30 min (all at 24 °C). Incubation with primary antibodies was performed overnight at 4 °C. Secondary antibodies were added the next day for 1.5–2 h at room temperature. The antibodies were diluted in PBS with 1% BSA, as indicated in Table 2. Nuclei were stained with DAPI. Preparations were analyzed using the Nikon Eclipse Ti-E microscope (Nikon, Tokyo, Japan) and NIS Elements software Advenced Research version 4.30.

### 4.9. Differentiation of iPSCs into Astrocytes

Differentiation into astrocytes was carried out according to the established protocol [16], with modifications. Briefly, iPSCs were passaged onto Matrigel-GFR (Corning, New York, NY, USA) in high density (80–90% confluency) and cultured for 24 h in Essential 8 medium (Thermo Fisher Scientific, Waltham, MA, USA). Next, the cells were cultured in the basic medium for neural differentiation (Neurobasal Medium—48.25%, F12/DMEM—48.25%, GlutaMAX—1%, Penicillin/Streptomycin—1%, N2—0.5%, B27 without vitamin A—1% (all from Thermo Fisher Scientific, Waltham, MA, USA) and ascorbic acid—0.2 mM (Sigma-Aldrich, Darmstadt, Germany)) with the addition of factors (Table 3) according to the protocol scheme (Figure 2A).

Dissociation of a dense monolayer of cells was performed using StemPro™ Accutase™ (Accutase, Thermo Fisher Scientific, Waltham, MA, USA) on day 11 with the addition of ROCK inhibitor for the first 24 h. Next, 20 ng/mL bFGF and EGF (Table 3) were added from day 13, and the cells were cultured with weekly passaging in a ratio of 1:2 or 1:3 using Accutase. The remaining cells were cryopreserved in a medium composed of 50% basic medium, 40% Fetal bovine serum (Thermo Fisher Scientific, Waltham, MA, USA), and 10% DMSO (Sigma-Aldrich, Darmstadt, Germany). CNTF (PeproTech, Cranbury, NJ, USA) was added to the culture medium for terminal differentiation and maturation of astrocytes after day 25.

## 5. Conclusions

Cell models provide an excellent opportunity to study the mechanisms of hereditary disease development. In the work, a cell model of PD was obtained. Two iPSC lines were generated from PBMCs of a patient carrying N370S mutation in the *GBA1* gene. The iPSCs express pluripotency markers and are able to differentiate into any type of cell. We performed a successful astroglial differentiation and obtained iPSC-derived astrocytes that were positive for specific markers GFAP and S100β. This cell model allows for studying the impact of astroglial dysfunction caused by mutations in the *GBA1* gene on the process of neurodegeneration. The astroglial cells obtained can enhance the cell model of PD by co-cultivating with iPSC-derived dopaminergic neurons or by generating 3D brain organoids. This provides a comprehensive tool to study the involvement of GCase in the molecular pathogenesis of PD and to test potential drugs.

## Figures and Tables

**Figure 1 ijms-25-00327-f001:**
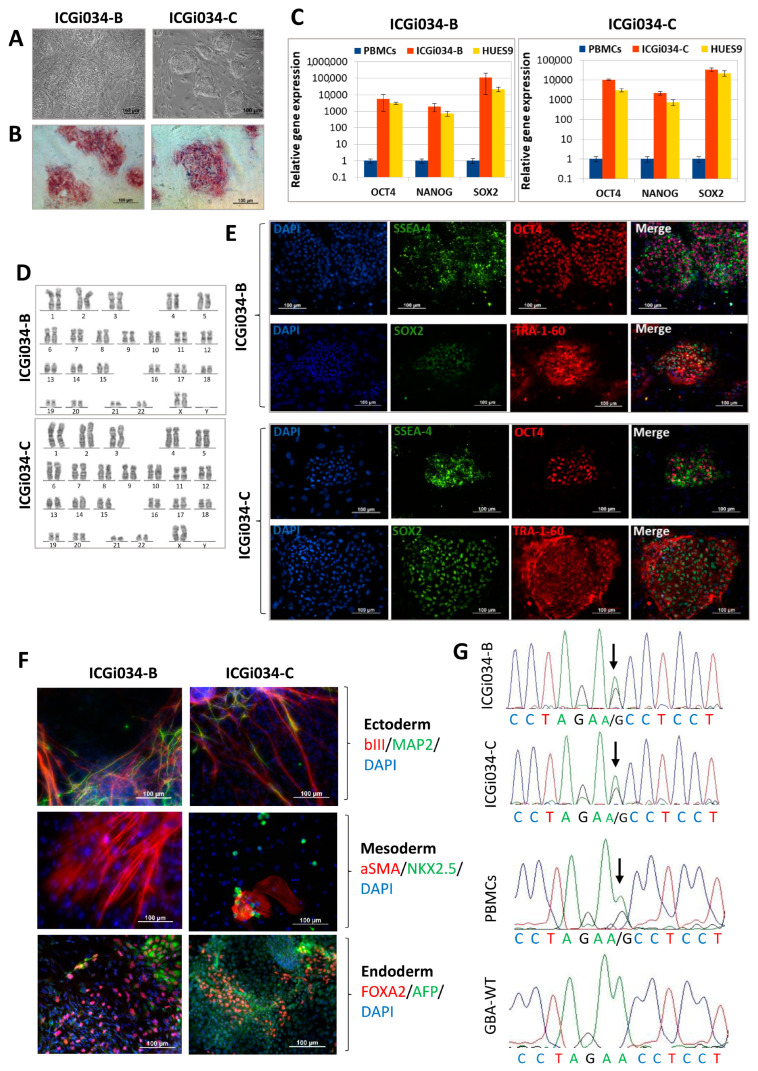
Characterization of the induced pluripotent stem cells (iPSC) cell lines ICGi034-B and ICGi034-C: (**A**) Morphology of iPSC colonies. (**B**) Quantitative analysis of the expression of pluripotency markers (NANOG, OCT4, SOX2) using RT-qPCR. Error bars show standard deviation. (**C**) Histochemical detection of alkaline phosphatase. (**D**) Karyotype analysis (G-banding) (46,XX). (**E**) Immunofluorescent staining for pluripotency markers OCT4, SOX2, SSEA-4, TRA-1-60. (**F**) Immunofluorescent staining for differentiation markers: αSMA (red signal) and NKX2.5 (green signal) (mesoderm); TUBB3 (red signal) and MAP2 (green signal) (ectoderm); FOXA2 (red signal) and AFP (green signal) (endoderm). Nuclei are stained with DAPI (blue signal). (**G**) Sequence images of the *GBA1* gene regions vof the patient’s peripheral blood mononuclear cells (PBMCs), iPSC lines ICGi034-B and ICGi034-C, and a healthy donor (GBA-WT). Identified polymorphisms are marked with arrows. All scale bars—100 μm.

**Figure 2 ijms-25-00327-f002:**
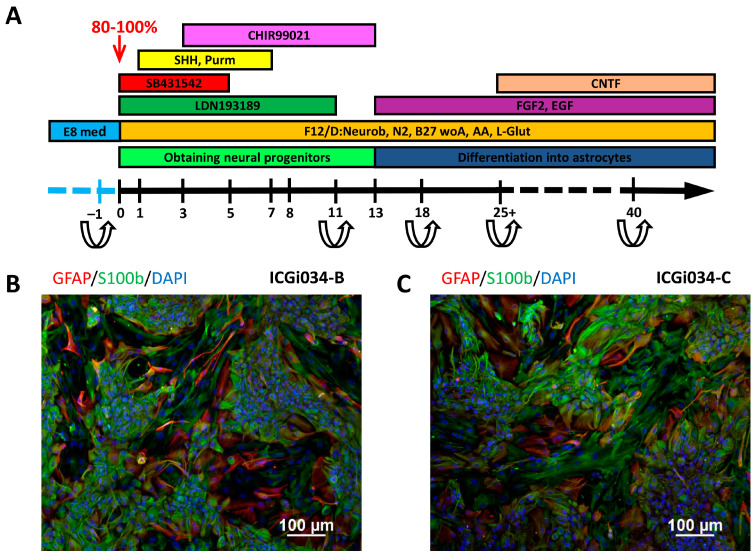
Differentiation of iPSC cell lines ICGi034-B and ICGi034-C: (**A**) Scheme of iPSC differentiation protocols into astrocytes. (**B**,**C**) Immunofluorescent staining for markers of astrocyte terminal differentiation GFAP and S100β.

**Table 1 ijms-25-00327-t001:** Characterization and validation of the new lines.

Classification	Test	Result	Data
Morphology	Photography Bright field	Normal	Figure 1A
Pluripotency status	Qualitative analysis: Alkaline phosphatase staining	Positive	Figure 1B
	Qualitative analysis: Immunocytochemistry	Positive staining for pluripotency markers: OCT3/4, SOX2, TRA-1-60, SSEA-4	Figure 1E
	Quantitative analysis: RT-qPCR	Expression of pluripotency markers: *NANOG*, *OCT4*, *SOX2*	Figure 1C
Genotype	Karyotype (G-banding)	46,XX	Figure 1D
Mutation analysis	Sanger sequencing of DNA from patient’s PBMCs and iPSCs	Heterozygous p.N370S in *GBA1*	Figure 1G
Differentiation potential	Embryoid body formation	Positive staining for germ layer markers: ɑSMA and NKX2.5 (mesoderm); MAP2 and TUBB3/TUJ1 (ectoderm); FOXA2/AFP (endoderm)	Figure 1F
Specific pathogen-free status	Mycoplasma	Negative	Appendix A

**Table 2 ijms-25-00327-t002:** Reagents details.

Antibodies Used for Immunocytochemistry			
	Antibody	Dilution	Company Cat # and RRID
Pluripotency Markers	Rabbit IgG anti-OCT4	1:200	Abcam, Cambridge, UK, Cat # ab18976, RRID:AB_444714
	Mouse IgG3 anti-SSEA4	1:200	Abcam, Cambridge, UK, Cat # ab16287, RRID:AB_778073
	Mouse IgM anti-TRA-1-60	1:200	Abcam, Cambridge, UK, Cat # ab16288, RRID:AB_778563
	Rabbit IgG anti-SOX2	1:500	Cell Signaling, Danvers, MA, USA, Cat # 3579, RRID:AB_2195767
Differentiation Markers	Mouse IgG2a anti-αSMA	1:100	Dako, Glostrup, Denmark, Cat # M0851, RRID:AB_2223500
	Rabbit IgG anti-NKX2.5 (H-114)	1:100	Santa Cruz Biotechnology, Dallas, TX, USA, Cat # sc-14033, RRID:AB_650281
	Mouse IgG2a anti-AFP	1:250	Sigma-Aldrich, Darmstadt, Germany, Cat # A8452, RRID:AB_258392
	Mouse IgG2a anti-Tubulin β 3 (TUBB3)/Clone: TUJ1	1:1000	BioLegend, San Diego, CA, USA, Cat # 801,201, RRID:AB_2313773
	Chicken IgG anti MAP2	1:1000	Abcam, Cambridge, UK, Cat # ab5392, RRID:AB_2138153
	Mouse IgG1 anti-HNF3b (FOXA2)	1:50	Santa Cruz Biotechnology, Dallas, TX, USA, Cat # sc-374,376, RRID:AB_10989742
	Mouse IgG1 anti-S100β	1:400	Sigma-Aldrich, Darmstadt, Germany, Cat # S2532, RRID:AB_477499
	Rabbit IgG anti-GFAP	1:200	Dako, Glostrup, Denmark, Cat # Z0334
Secondary antibodies	Goat anti-Mouse IgG (H + L) Secondary Antibody, Alexa Fluor 488	1:400	Thermo Fisher Scientific, Waltham, MA, USA, Cat # A11029, RRID:AB_2534088
	Goat anti-Mouse IgG (H + L) Secondary Antibody, Alexa Fluor 568	1:400	Thermo Fisher Scientific, Waltham, MA, USA, Cat # A11031, RRID:AB_144696
	Goat anti-Rabbit IgG (H + L) Highly Cross-Adsorbed Secondary Antibody, Alexa Fluor 488	1:400	Thermo Fisher Scientific, Waltham, MA, USA, Cat # A11008, RRID:AB_143165
	Goat anti-Rabbit IgG (H + L) Alexa Fluor 568	1:400	Thermo Fisher Scientific, Waltham, MA, USA, Cat # A11011, RRID:AB_143157
	Goat anti-Mouse IgG1 Alexa Fluor 568	1:400	Thermo Fisher Scientific, Waltham, MA, USA, Cat # A21124, RRID:AB_2535766
	Goat anti-Mouse IgG2a Alexa Fluor 488	1:400	Thermo Fisher Scientific, Waltham, MA, USA, Cat # A21131, RRID:AB_2535771
	Goat anti-Chicken IgY (H + L) Alexa Fluor 488	1:400	Abcam, Cambridge, UK, Cat # ab150173, RRID:AB_2827653
Primers			
	Target	Size of band	Forward/Reverse primer (5′-3′)
Episomal plasmid vector detection	EBNA-1	61 bp	TTCCACGAGGGTAGTGAACC/ TCGGGGGTGTTAGAGACAAC
Mycoplasma detection	16S ribosomal RNA gene	280 bp	GGGAGCAAACAGGATTAGATACCCT/ TGCACCATCTGTCACTCTGTTAACCTC
House-keeping gene (RT-qPCR)	beta-2-microglobulin	280 bp	TAGCTGTGCTCGCGCTACT/ TCTCTGCTGGATGACGTGAG
Pluripotency marker (RT-qPCR)	*NANOG*	116 bp	TTTGTGGGCCTGAAGAAAACT/ AGGGCTGTCCTGAATAAGCAG
	*OCT4*	94 bp	CTTCTGCTTCAGGAGCTTGG/ GAAGGAGAAGCTGGAGCAAA
	*SOX2*	100 bp	GCTTAGCCTCGTCGATGAAC/ AACCCCAAGATGCACAACTC
Targeted mutation analysis	*GBA1*	600 bp	CTGTTGCTACCTAGTCACTTCC/ CCCTATCTTCCCTTTCCTTCAC

**Table 3 ijms-25-00327-t003:** List of used growth factors, inhibitors and small molecules for astrocyte differentiation.

Substance	Company	Concentration	Days
LDN193189	Sigma-Aldrich, Darmstadt, Germany	100 nM	0–11
SB431542	Abcam, Cambridge, UK	10 µM	0–5
Purmorphamine	Tocris, Ellisville, MO, USA	2 µM	1–7
SHH C25II	PeproTech, Cranbury, NJ, USA	100 ng/mL	1–7
CHIR99021	Sigma-Aldrich, Darmstadt, Germany	3 µM	3–13
bFGF	SCI Store, Moscow, Russia	20 ng/mL	13–…
EGF	PeproTech, Cranbury, NJ, USA	20 ng/mL	13–…
CNTF	PeproTech, Cranbury, NJ, USA	10 ng/mL	25–…

## Data Availability

The data presented in this study are openly available in the Human Pluripotent Stem Cell Registry (https://hpscreg.eu/cell-line/ICGi034-B and https://hpscreg.eu/cell-line/ICGi034-C, accessed on 1 December 2023).

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
