# Peer review of "IPSC-Derived Astrocytes Contribute to In Vitro Modeling of Parkinson’s Disease Caused by the GBA1 N370S Mutation"

_ijms, 2023, doi:10.3390/ijms25010327_

Round 1

Reviewer 1 Report

Comments and Suggestions for Authors

The manuscript titled " Cell Model of Parkinson's Disease Caused by a Mutation in the GBA1 Gene" focuses on the development of a cell model for Parkinson's disease (PD) associated with a mutation in the GBA1 gene. The study generated two induced pluripotent stem cell (iPSC) lines from a PD patient with a GBA1 mutation and differentiated them into astrocytes, aiming to investigate the role of astrocytes in GBA-associated PD pathogenesis. The iPSC lines can also be differentiated into dopaminergic neurons, offering a comprehensive tool for studying the molecular pathogenesis of Parkinson's disease and testing potential drugs. While the manuscript is informative, there are suggestions for improvement,

1.       The title is informative, clearly indicating the focus of the manuscript on a cell model of Parkinson's disease caused by a mutation in the GBA1 gene. However, it might be improved by specifying the type of cell model and highlighting its significance or novelty.

2.       The abstract provides a concise overview of the study, introducing the importance of creating suitable models for understanding Parkinson's disease (PD) pathogenesis. However, the abstract lacks specific details about the methods used, key findings, and potential implications of the study. Consider adding information on the significance of the generated cell model and how it contributes to current understanding or therapeutic advancements in PD.

3.       The introduction effectively introduces Parkinson's disease and the relevance of the GBA1 gene mutation. The introduction lacks a clear statement of the study's objectives and hypotheses. Clearly stating these elements would provide readers with a roadmap for the rest of the manuscript. It is advisable to provide more recent references, as the field of Parkinson's disease is rapidly evolving.

4.       The discussion effectively links the results to the broader context of Parkinson's disease and emphasizes the role of astrocytes. However, the discussion could benefit from a more critical analysis of the limitations of the study and potential confounding factors. Consider discussing how the findings compare with existing literature and whether any unexpected results were encountered.

5.       The methods section is detailed and provides clarity on the procedures followed. Consider including more information on statistical analyses, particularly if used in data interpretation.  Provide details on the rationale behind choosing specific methods and highlight any modifications made to established protocols.

6.       The conclusion summarizes the main findings and emphasizes the significance of the cell model. Consider providing explicit recommendations for future research directions based on the current study.

Comments on the Quality of English Language

The manuscript would benefit from careful proofreading for grammatical errors and clarity.

Author Response

Dear reviewer,

Thank you for reviewing our Brief Report. We have carefully considered your comments and made the necessary corrections to the manuscript.

  1. The title is informative, clearly indicating the focus of the manuscript on a cell model of Parkinson's disease caused by a mutation in the GBA1 gene. However, it might be improved by specifying the type of cell model and highlighting its significance or novelty.

The title has been adjusted accordingly.

New title: “IPSC-derived astrocytes contribute to in vitro modeling of Parkinson's disease caused by the GBA1 N370S mutation”

  1. The abstract provides a concise overview of the study, introducing the importance of creating suitable models for understanding Parkinson's disease (PD) pathogenesis. However, the abstract lacks specific details about the methods used, key findings, and potential implications of the study. Consider adding information on the significance of the generated cell model and how it contributes to current understanding or therapeutic advancements in PD.

      We have made appropriate changes to the text. Clarifications of methods used, key findings, and potential implications of the study have been added to the abstract.

  1. The introduction effectively introduces Parkinson's disease and the relevance of the GBA1 gene mutation. The introduction lacks a clear statement of the study's objectives and hypotheses. Clearly stating these elements would provide readers with a roadmap for the rest of the manuscript. It is advisable to provide more recent references, as the field of Parkinson's disease is rapidly evolving.

We have revised the introduction and formulated the objectives of the study.

  1. The discussion effectively links the results to the broader context of Parkinson's disease and emphasizes the role of astrocytes. However, the discussion could benefit from a more critical analysis of the limitations of the study and potential confounding factors. Consider discussing how the findings compare with existing literature and whether any unexpected results were encountered.

We have made appropriate changes to the text.

  1. The methods section is detailed and provides clarity on the procedures followed. Consider including more information on statistical analyses, particularly if used in data interpretation. Provide details on the rationale behind choosing specific methods and highlight any modifications made to established protocols.

      Thank you for your comment. The real-time PCR graphs were analyzed using statistical methods. Corresponding statements have been added to the 'Materials and Methods' section and the caption of Figure 1.

  1. The conclusion summarizes the main findings and emphasizes the significance of the cell model. Consider providing explicit recommendations for future research directions based on the current study.

We have revised the conclusions in accordance with the recommendation.

Reviewer 2 Report

Comments and Suggestions for Authors

Here, these workers have reprogrammed PBMCs from a patient carrying the high risk Parkinson’s disease (PD) GBA1 gene mutation c.1226A>G (p.N370S) and generated two iPSC lines, ICGi034-B and ICGi034-C also carrying this gene mutation. The iPSCs were then differentiated via neural stem cells into astrocytes expressing GFAP. It is proposed that the resulting cell model can be utilised for studying the involvement of astrocytes in the pathogenesis of GBA-associated PD.

I am not  a molecular biologist but the methodology used to generate the production of astrocytes from PBMCs carrying the GBA mutation looked convincing to me and each step has been validated. My question as a clinical neuroscientist would be why have these workers targeted astrocyte rather than neurone production from the mutant neuronal stem cells they first produce? It is generally thought that pathological GBA mutations act by reducing the ability of GCase to help clear α-synuclein aggregates taken up by neuronal lysosomes. This is in contrast to multiple system atrophy (MSA), where glial α-synuclein aggregates are widespread, whereas in PD they are found in neuronal Lewy bodies and in neurites. There is no doubt that astrocyte activation and dysfunction does occur in PD but it is likely to be secondary to the neuronal pathology rather than be causative. It seems these workers may have actually created a cell model more relevant to MSA than PD though I am not aware that GBA mutations are associated with MSA. Researchers interested in that particular atypical parkinsonian condition should be alerted to this work.

Author Response

Dear reviewer,

Thank you for your interest in our work. We have made some changes to the text of the article based on your comments.

Comments and Suggestions for Authors

Here, these workers have reprogrammed PBMCs from a patient carrying the high risk Parkinson’s disease (PD) GBA1 gene mutation c.1226A>G (p.N370S) and generated two iPSC lines, ICGi034-B and ICGi034-C also carrying this gene mutation. The iPSCs were then differentiated via neural stem cells into astrocytes expressing GFAP. It is proposed that the resulting cell model can be utilised for studying the involvement of astrocytes in the pathogenesis of GBA-associated PD.

My question as a clinical neuroscientist would be why have these workers targeted astrocyte rather than neuron production from the mutant neuronal stem cells they first produce? It is generally thought that pathological GBA mutations act by reducing the ability of GCase to help clear α-synuclein aggregates taken up by neuronal lysosomes. This is in contrast to multiple system atrophy (MSA), where glial α-synuclein aggregates are widespread, whereas in PD they are found in neuronal Lewy bodies and in neurites. There is no doubt that astrocyte activation and dysfunction does occur in PD but it is likely to be secondary to the neuronal pathology rather than be causative. It seems these workers may have actually created a cell model more relevant to MSA than PD though I am not aware that GBA mutations are associated with MSA. Researchers interested in that particular atypical parkinsonian condition should be alerted to this work.

We appreciate your valuable comment. Indeed, dopaminergic neurons are the main point of interest for researchers studying PD mechanisms. Recently, we developed a cell model based on iPSC-derived dopaminergic neurons (Grigor'eva et al., 2023, doi: 10.3390/ijms24054437) and showed a decrease in GCase activity and dysregulation of other lysosomal enzymes in dopaminergic neurons differentiated from iPSCs of patients with GBA1 mutations. We also used the cell model to study the efficacy of molecular chaperones targeting mutant GCase (Grigor'eva et al., 2023, doi: 10.3390/ijms24054437, Kopytova et al., 2023, doi: 10.3390/ijms24109105).

However, cell culture of iPSC-derived neurons has many limitations as a cell model of PD. In the brain, all cell types are in close interaction and communication, and all brain cell types are involved in disease pathogenesis. It is important to improve the in vitro cell models to be more relevant. One way of improvement is the co-culture of neurons and astrocytes in both 2D and 3D models (in brain organoids).

We have added this idea in the text of the article.